# The Clinical and Economic Impact of Antibiotic Resistance in China: A Systematic Review and Meta-Analysis

**DOI:** 10.3390/antibiotics8030115

**Published:** 2019-08-10

**Authors:** Xuemei Zhen, Cecilia Stålsby Lundborg, Xueshan Sun, Xiaoqian Hu, Hengjin Dong

**Affiliations:** 1Center for Health Policy Studies, School of Public Health, Zhejiang University School of Medicine, Hangzhou 310058, China; 2Global Health-Health Systems and Policy (HSP), Medicines, Focusing Antibiotics, Department of Public Health Sciences, Karolinska Institutet, 17177 Stockholm, Sweden

**Keywords:** antibiotic resistance, multi-drug resistance, mortality, hospital stay, hospital cost, China

## Abstract

Antibiotic resistance (ABR) is one of the biggest threats to global health, especially in China. This study aims to analyze the published literature on the clinical and economic impact of ABR or multi-drug resistant (MDR) bacteria compared to susceptible bacteria or non-infection, in mainland China. English and Chinese databases were searched to identify relevant studies evaluating mortality, hospital stay, and hospital costs of ABR. A meta-analysis of mortality was performed using a random effects model. The costs were converted into 2015 United States (US) dollars. Of 13,693 studies identified, 44 eligible studies were included. Twenty-nine investigated the impact of ABR on hospital mortality, 37 were focused on hospital stay, and 21 on hospital costs. Patients with ABR were associated with a greater risk of overall mortality compared to those with susceptibility or those without infection (odds ratio: 2.67 and 3.29, 95% confidence interval: 2.18–3.26 and 1.71–6.33, *p* < 0.001 and *p* < 0.001, respectively). The extra mean total hospital stay and total hospital cost were reported, ranging from 3 to 46 days, and from US$238 to US$16,496, respectively. Our study indicates that ABR is associated with significantly higher mortality. Moreover, ABR is not always, but usually, associated with significantly longer hospital stay and higher hospital costs.

## 1. Introduction

Antibiotic use and misuse has led to the emergence and development of antibiotic resistance (ABR), which is one of the biggest threats to global public health [1]. This problem is particularly acute in China because of antibiotic prescribing behavior, including: inappropriate financial incentives, over-the-counter availability of antibiotics, and the widespread antibiotic use and misuse in agriculture [2]. BRICS countries (Brazil, Russia, India, China and South Africa) have shown the highest rates of antibiotic use, accounting for 76% of the overall increase in global antibiotic consumption between the years 2000 and 2010. Up to 57% of the increase in the hospital sector was attributable to China [3,4]. China was the second largest consumer of antibiotics in 2010. Meanwhile, China has high prescription rates of antibiotics for both inpatients and outpatients [5]. There is also a high use of injections in China, with about one-third of the prescriptions for injections being written in community health institutions. This rate is two to three times higher than the World Health Organization (WHO) standard and estimates from other developing countries [6].

Consequently, as a result of this antibiotic misuse, China has the highest level of ABR and the most rapid growth of ABR globally [7,8]. Data from the 2017 China Antimicrobial Resistance Surveillance System showed that the national rates of methicillin-resistant *Staphylococcus aureus* (MRSA), third-generation cephalosporin-resistant *Escherichia coli*, carbapenem-resistant *Klebsiella pneumoniae* (CRKP), third-generation cephalosporin-resistant *K. pneumoniae*, carbapenem-resistant *Pseudomonas aeruginosa* (CRPA), and carbapenem-resistant *Acinetobacter baumannii* (CRAB) were 32.2%, 54.2%, 9.0%, 33.0%, 20.7%, and 56.1%, respectively [9], and there were regional differences across provinces in China [10]. The report from CHINET surveillance system of bacterial resistance showed that between 2005 and 2014, carbapenem resistance among *K. pneumoniae* isolates increased from 2.4% to 13.4%, and CRAB isolates increased from 31% to 66.7% [11].

To combat this trend, the Chinese government announced a national action plan to combat antimicrobial resistance in 2016 [12] in response to the global action plan by WHO [13]. On 1 July 2011, the Chinese government carried out a three-year national public hospital campaign targeting ABR [14,15]. This action plan, as a combination of managerial and professional strategies, was effective in reducing antibiotic prescribing rates and intensity of antibiotic consumption. On 1 August 2012, the Chinese government formally implemented administrative regulations for the clinical use of antibacterial agents [16]. In addition, China has built multi-disciplinary collaborations with the European Union, Sweden, the Netherlands, and the United Kingdom to stop the increasing the burden caused by ABR [14]. Even so, we still face a great challenge in controlling antibiotic use and antibiotic resistance in China.

ABR, especially multi-drug resistance (MDR), is associated with high mortality, increased resource utilization, and additional economic costs [17,18,19,20,21]. It is estimated that 1 million deaths will be attributed to antimicrobial resistance by 2050, and United States (US)$20 trillion cumulative costs will be lost if substantive efforts are not made to tackle this problem [22].

Despite the evidenced threat posed by ABR, information on its clinical and economic impact is limited in China. Assessments of the burden of ABR is a key step for the implementation of national strategies to combat ABR, so we can clearly know the costs and benefits of national action plans [13]. However, there has not been a contemporary literature review reporting on the clinical and economic impact of ABR in mainland China. In this study, we aimed to analyze the published literature of the clinical and economic consequences of ABR or MDR bacteria compared to susceptible bacteria and uninfected individuals. We also conducted a meta-analysis of hospital mortality to quality the impact of ABR or MDR on clinical outcomes.

## 2. Materials and Methods

### 2.1. Literature Search

A systematic search in the English databases (PubMed, Web of Science, and Embase) and Chinese databases (China National Knowledge Infrastructure, Wanfang data, and Chongqing VIP) up to 16 January 2019, was carried out. In addition, manual reference checks from retrieved studies were performed to ensure inclusion of all relevant studies. Detailed search strategies are provided in Appendix A.

### 2.2. Study Selection

Inclusion criteria were (1) studies published in English or Chinese language; (2) publication date between 1 January 2000 and 16 January 2019; (3) original research using any study designs, such as cohort, case–control, or observational studies; (4) reports on humans; (5) reports in China; (6) reports on resistant versus susceptible cases; and (7) reports on clinical and economic outcomes. In order to ensure the analysis focuses on contemporary literature that reflects current resistance patterns and clinical practice guidelines, studies published before 2000 were not considered [20,23]. Two reviewers (XZ, XS) independently reviewed titles and abstracts, then assessed the full-text to decide whether it met the inclusion criteria. Disagreements were resolved by a third reviewer (XH).

### 2.3. Data Extraction

The extracted data included first author, publication year, type of study, method, province, hospital setting, study period, study population, types of infection, hospital ward, organisms, and sample size (cases and controls). The following outcomes were extracted: all-cause mortality, attributable mortality, 30-day (28-day) mortality, crude mortality; total hospital stay, length of stay before/after infection, intensive care unit (ICU) stay; and total hospital costs/charges, hospital costs/charges before/after infection, and antibiotic costs. All presented *p*-values were obtained from analyses within the included studies. MDR was defined that if it is resistance to three or more than three types of antibiotics or if the isolated bacteria were MDR organisms, such as MRSA, CRPA, and CRAB. In addition, both intermediate and resistant isolates were regarded as “resistant”.

### 2.4. Study Quality Assessment

We assessed the included study quality using the Newcastle-Ottawa quality assessment Scale (NOS) for cohort and case–control studies. The NOS includes four domains and nine “stars”, where >6 stars indicates high-quality studies, 4–6 stars indicates moderate quality, and ≤3 stars indicates low quality [20,23,24] (Appendix A).

### 2.5. Data Analysis

Meta-analysis was conducted to determine overall mortality associated with ABR or MDR. Sub-group analyses for mortality were performed based on bacteria and three economic zones in China where there were three or more studies that could be combined. Heterogeneity was calculated as *I*^2^ statistic values, which were categorized as low (0–50%), moderate (50–75%), or high (above 75%). All values were calculated with 95% confidence intervals (CI), and the results presented as odds ratios (OR). For other outcomes, a meta-analysis was not possible due to a variety of study designs and reporting values (mean or median). Costs were converted into 2015 US dollars by annual consumer price index and 2015 average exchange rates [25,26].

## 3. Results

### 3.1. Study Identification

A total of 13,693 studies were identified from the searches. One study was added following a hand search of the references of included studies. Of these, 8579 studies were excluded because they did not fulfill the inclusion criteria based on their title and abstracts after excluding duplicates (4770 studies). For the remaining 345 studies, we screened full texts and identified 44 potentially relevant studies (Figure 1).

### 3.2. Study Characteristics and Quality

Of the 44 eligible studies included in our review, 29 studies investigated the impact of ABR on mortality (Table 1, Appendix A), 37 studies reported on hospital length of stay or ICU stay (Table 2, Appendix A), and 21 studies reported on economic consequences (Table 3, Appendix A). The majority of studies were retrospective observational studies (43 studies) and were conducted in a single hospital setting (39 studies). The study data were obtained from 17 provinces in mainland China, with the largest number of studies from Zhejiang province (*n* = 10), followed by Shanghai (*n* = 6), Beijing (*n* = 5), Hubei (*n* = 5), Sichuan (*n* = 4), Chongqing (*n* = 3), Guangdong (*n* = 3), Jiangsu (*n* = 3), Shandong (*n* = 3), Anhui (*n* = 2), Fujian (*n* = 2), Hebei (*n* = 1), Hunan (*n* = 1), Yunnan (*n* = 1), and Ningxia (*n* = 1) (Figure 2). The majority of studies (*n* = 20) collected data from the eastern economic zone, and only four studies were from the central economic zone and western economic zone, respectively. Those economic zones were divided according to geographical location and economic development in China (Figure 2). Most of the studies (*n* = 13) reported on a group of bacteria, 11 on *A. baumannii*, 8 on *K. pneumoniae*, 8 on *S. aureus*, 5 on *P. aeruginosa*, 2 on *E. coli*, 1 on *Enterococcus*, 1 on coagulase-negative *Staphylococci*, and 1 on *Proteus mirabilis*. Statistical tests were the most utilized analysis method in the included studies, and propensity score matching, simple matching, regression model, and generalized linear model were conducted to control for baseline characteristics (Appendix A). Regarding the quality of included studies, 16 were high quality and 28 were moderate quality (Appendix A).

### 3.3. Mortality Outcome 

A total of 29 studies reported data on hospital mortality [27,28,29,30,31,32,33,34,35,36,37,38,39,40,41,42,43,44,45,46,47,48,49,50,51,52,53,54,55]. We found ABR had a significant impact on mortality in 22 studies (Table 1, Appendix A). Three studies included two different comparisons based on different study designs [38,43,51], two different comparisons contributed data for both susceptible bacterial infections and those without infections in two studies [45,55], and two other studies contained two and four different descriptions of mortality (attributable or all-cause in hospital mortality/28-day (30-day) hospital mortality) [44,54]. Patients with infections due to ABR or MDR bacteria had a higher odds of overall mortality than those patients with susceptible bacterial infections or control patients without infection (OR: 2.67, 95% CI: 2.18–3.26, *p* = 0.001; OR: 3.29, 95% CI: 1.71–6.33, *p* = 0.001) with moderate heterogeneity (*I^2^* = 52.5%, *P* <0.001) (Figure 3A), and low heterogeneity (*I^2^* = 52.5%, *p* <0.001), respectively (Figure 3B). A high risk of mortality due to ABR or MDR *P. aeruginosa* was observed with high statistical significance (OR: 3.38, 95% CI: 1.81–6.31, *p* <0.001) with moderate heterogeneity (*I^2^* = 57.9%, *p* = 0.050), followed by gram-negative bacteria (OR: 3.30, 95% CI: 1.56–6.97, *p* = 0.002) with moderate heterogeneity (*I^2^* = 54.9%, *p* = 0.109), *K. pneumoniae* (OR: 3.12, 95% CI: 1.99–4.89, *p* <0.001) with moderate heterogeneity (*I^2^* = 73.9%, *p* <0.001), *A. baumannii* (OR: 2.18, 95% CI: 1.70–2.80, *p* <0.001) with low heterogeneity (*I^2^* = 0.0%, *p* = 0.453), and *S. aureus* (OR: 1.55, 95% CI: 0.95–2.53, *p* = 0.082) with low heterogeneity (*I^2^* = 0.0%, *p* = 0.482) (Figure 3C). High statistical significance was observed in the central economic zone (OR: 5.14, 95% CI: 1.80–14.70, *p* = 0.002) with moderate heterogeneity (*I^2^* = 51.6%, *p* = 0.103), the eastern economic zone (OR: 2.74, 95% CI: 2.12–3.55, *p* <0.001) with moderate heterogeneity (*I^2^* = 57.1%, *p* <0.001), and the western economic zone (OR: 2.14, 95% CI: 1.39–3.27, *p* = 0.001) with low heterogeneity (*I^2^* = 45.2%, *p* = 0.121) (Figure 3D). 

### 3.4. Hospital Stay Outcome

A total of 37 studies reported data on hospital stay [27,28,29,31,32,33,34,35,36,37,38,39,40,41,43,48,49,50,51,52,54,55,56,57,58,59,60,61,62,63,64,65,66,67,68,69,70]. It is difficult to directly compare length of stay across eligible studies due to different definitions and measures (mean or median). The extra mean total length of stay ranged from 3 days between MDR gram-negative/gram-positive community-acquired infection and susceptible cases [58], to 46 days between MDR *P. aeruginosa* and non-MDR cases [65]. An extra median total length of stay was observed ranging from 4 days between CRPA and CSPA groups [43] to 26 days between CRKP bloodstream infection and carbapenem-susceptible *K. pneumoniae* (CSKP) groups [43].

Two studies reported that compared with patients without an infection, patients with MRSA infections were associated with extra median total length of stay of 14 days [56] and extra mean total length of stay of 9 days after adjusting for confounders [36]. Compared with patients with methicillin-susceptible *S. aureus* (MSSA), MRSA cases were associated with longer median or mean total length of stay, length of stay before infection, and length of stay after infection in univariate analyses in most studies [37,38,65,66]; however, there was no significant difference between the two groups after controlling for baseline factors [38]. We also found there was no significant difference between the two groups in univariate analyses in some studies [37,41,49]. For other gram-positive bacteria (coagulase-negative *Staphylococci* [65] and *Enterococcus* [54]), there was a significant difference in hospital length of stay between resistant or MDR groups and susceptible, non-MDR, or non-infected groups in univariate analyses (Table 2, Appendix A). 

Among patients with *E. coli* and *K. pneumoniae* intra-abdominal infection (IAI), one study reported longer mean total length of stay between extended spectrum βlactamases (ESBL)-positive and ESBL-negative groups (24 days vs. 15 days) in a generalized linear model [67]. For only *K. pneumoniae* or only *E. coli*, we found significant differences in total length of stay between CRKP and non-CRKP [29], between resistant enzyme-producing and non-resistant enzyme-producing [59], between MDR and non-MDR [65], and between CRKP and CSKP [49,51] in univariate analyses, even after propensity score matching for potential confounding variables [51]. We also found a significant relationship in ICU length of stay between CRKP and non-CRKP groups, and significant impacts on length of stay before infection between resistant enzyme-producing and non-resistant enzyme-producing groups and between CRKP and CSKP groups [50,51,59]. However, there was no significant difference in total length of stay or length of stay after infection between CRKP and CSKP groups in univariate analyses [48,49,50,51] (Table 2 and Appendix A). 

For *A. baumannii*, patients with MDR infections were associated with significantly longer mean total length of stay than non-MDR cases, ranging from 5 days for children to 13 days for adults [27,31,33,55,63,65]. One study reported a 6-day additional median total length of stay for CRAB vs. carbapenem-susceptible *A. baumannii* (CSAB) [39]. Patients with MDR *A. baumannii* or CRAB had a greater ICU length of stay than those with non-MDR *A. baumannii* [55] or CSAB [39], respectively. However, there was no significant difference in length of stay before infection between CRAB and CSAB groups in univariate analyses in some studies [39,49,68]. Three studies reported length of stay among patients with *P. aeruginosa* [32,43,65], and we found that carbapenem resistance was associated with significant impacts on total length of stay, length of stay after infection, and length of stay admitting the ICU. For patients with gram-negative and gram-positive bacteria, resistance or MDR was associated with significantly longer total length of stay or infection related length of stay than non-resistant, non-MDR, or non-infection patients [52,57,58,60,61,64,69]. In addition, patients with carbapenem-resistant or MDR gram-negative bacteria were associated with longer hospital length of stay than those with carbapenem-susceptible or non-MDR gram-negative bacteria in two studies [34,62]; however, we also found there was no significant difference in total length of stay or ICU length of stay among children with non-fermenters sepsis even after adjustment for baseline variables [40] (Table 2, Appendix A).

### 3.5. Hospital Cost Outcomes

A total of 21 studies reported outcomes related to hospital costs or charges [27,28,38,39,40,43,44,51,52,56,57,58,60,61,63,64,65,67,68,69,70]. Additional mean total hospital costs ranged from US$238 among patients with ESBL-positive *E. coli*/*K. pneumoniae* IAI versus ESBL-negative cases [67], to US$16,496 among patients with ABR gram-positive/gram-negative bacteria versus uninfected cases [60], and additional mean antibiotic costs ranged from US$58 among patients with ESBL-positive *E. coli*/*K. pneumoniae* IAI versus ESBL-negative cases [67] to US$3240 among patients with MDR IAI versus non-MDR cases [69] (Table 3, Appendix A). 

The median total hospital cost was US$15,763 for MRSA cases and US$2185 for uninfected patients, accounting for an excess cost of US$13,578 attributable to MRSA after matching on relevant variables [56], however, there was no significant difference between MRSA and MSSA groups in two studies, whether or not they adjusting for risk factors [38,70]. ESBL-positive *E. coli* or/and *K. pneumoniae* patients incurred higher total hospital costs (US$541 vs. US$303) and antibiotic costs ($98 vs. US$40) [67]. Carbapenem-resistant *Escherichia coli* (CREC); was attributable to an extra total hospital cost of US$2380 and US$9851, compared with carbapenem-susceptible *E. coli* and uninfected groups, respectively [44]. After propensity score matching, patients with CRKP had higher hospital ($21,170 vs. US$11,313) and antibiotic costs ($2253 vs. US$1251) than those with CSKP during the entire hospitalization and during the period after infection (US$8912 vs. US$6677; US$973 vs. US$573) [51]. Patients with CRPA had a significantly higher total hospital cost and daily hospital cost than patients with CSPA in both unadjusted analysis and propensity score matching analysis [43]. Carbapenem resistance or MDR was significantly associated with higher total hospital cost and total antibiotic cost among patients with *A. baumannii* after accounting for confounding factors [27,39,63,68]. In addition, patients with resistant or MDR gram-negative and/or gram-positive bacteria were associated with higher total hospital costs and antibiotic costs than those with susceptible, non-MDR, or uninfected cases in most of studies [28,40,52,57,60,61,64,69]; however, there was no significant difference in total hospital cost between MDR gram-negative bacteria and non-MDR gram-negative bacteria in a univariate analysis in one study [28]. One study found that patients with MDR *E. coli*, *K. pneumoniae*, *Proteus mirabilis*, *A. baumannii*, *P. aeruginosa*, *Enterobacter cloacae*, *S. aureus*, or coagulase-negative *Staphylococci* were associated with significantly higher total hospital costs than non-MDR cases in univariate analyses [65] (Table 3, Appendix A).

## 4. Discussion

ABR is a global health crisis, especially in China, with high prescription rates for antibiotics in both inpatients and outpatients coupled with the highest level growth of ABR globally [8]. To our knowledge, this is the first systematic review to analyze the clinical and economic impact of ABR in China. It provides a clear picture of the real-world clinical and economic outcomes among patients with ABR, especially MDR, for clinicians, patients and researchers by merging information from both Chinese and English studies.

ABR and MDR are associated with significantly increased overall mortality as compared with susceptibility and non-infection (OR: 2.67, 95% CI: 2.18–3.26, *p* = 0.001; OR: 3.29, 95% CI: 1.71–6.33, *p* = 0.001, respectively), based on the pooled crude effect estimate, even though we found there was no significant difference between ABR or MDR and mortality in some studies, which is consistent with several studies in high-income, middle-income, and low-income countries [18,20,23,71,72,73,74,75,76]. This result may be overestimated because of the fact that most of patients with ABR, especially MDR, present with other mortality risk factors such as: severe illness, prolonged stay, ICU admission, invasive devices, and inappropriate antibiotic treatment. Therefore, this finding should be interpreted with caution as we did not adjust for such potential confounding factors. 

We suggest that ABR is not always, but usually, associated with significantly longer length of stay and higher hospital costs, which is consistent with other review studies [18,20,23]. Some studies may have lacked sufficient statistical power to detect significant differences in hospital stay and hospital costs. We found that a large number of studies addressing hospital stay or hospital costs calculated the mean or median values for different groups and performed univariate comparisons, with the results for different groups being more conservative after controlling baseline factors than univariate comparisons [38,43,51]; therefore, these results need to be interpreted with caution. Propensity scoring matching, simple matching, and multivariate analysis were the common methods used by studies to reduce the impact of potential confounding [27,36,38,39,40,43,51,56,57,60,63,68]. Some studies reported that ICU stay was associated with MDR [49,51]. The airborne and contact transmission of ABR bacteria in the ICU may result in healthcare-acquired infections among patients admitted to the ICU, especially for critically ill or immunocompromised patients who are associated with prolonged ICU stays, more invasive procedures, and greater exposure to more broad spectrum antibiotics [48,50]. This in turn likely contributes to higher mortality [50], longer hospital stay, and higher hospital costs [67]. These consequences further increase the likelihood of the spread of MDR bacteria. 

There were vast differences in both clinical and economic outcomes in different studies, which may be related to differences in consumption of classes of antibiotics, resistance patterns, and implementation of antibiotic stewardship programs in different provinces. China is extensive, with rich resources, and there are large differences in terms of the natural environment, socio-economic conditions, medical resources, medical conditions, health consciousness, and habits of medical treatment in different provinces. It is required that health authorities of different provinces develop antibiotic lists that meet local conditions [77]. Research with large sample sizes and multiple hospital settings on a national level and regional level is needed in the future in order to provide information for implementation of regional or national strategies for the containment of ABR, and to make a contribution to the global action plan on ABR. The report from the 2017 China Antimicrobial Resistance Surveillance showed that there were various differences in the morbidity from ABR or MDR in different provinces across mainland China. The detection rate of MRSA ranged from 16.6% in Shanxi province to 52% in the Tibet autonomous region. The resistance rate of CREC, third-generation cephalosporin-resistant *K. pneumoniae*, CRPA, and CRAB ranged from 0.3% in the Tibet autonomous region to 2.8% in Liaoning province, from 14.1% in Qinghai province to 53.8% in Henan province, from 8.7% in Ningxia Hui Autonomous Region to 30.2% in Liaoning province, and from 23.3% in Qinghai province to 80.4% in Henan province, respectively [10]. Therefore, the clinical and economic outcomes of ABR in different provinces, especially those that were not referred to in this study, but associated with high resistance rate, should attract the attention of researchers. *Enterococcus* and *E. coli*, defined as priority ABR bacteria by the WHO should gain further attention in China [10]. Methodological choices, description of values, target bacteria, and comparison groups can also lead to extreme variations in clinical and economic outcomes which studies reported.

In addition, there was geographical heterogeneity of studies reporting on clinical and economic outcomes in China. The most studies are limited chiefly to eastern economic zone, which is the most developed zone in China. Its consistent with the situation that similar analyses are needed for low- and middle- income countries [13]. The current status of ABR or MDR may be more serious in central and western economic zone because of lack of new medicines, diagnostic tools, and interventions. Moreover, compared with eastern economic zone, ABR or MDR in central and western economic zone may be associated with a higher mortality rate and higher economic burden, and a greater likelihood extreme poverty [78]; thus, the overall clinical and economic burden of ABR or MDR in China may be underestimated.

Our study has several limitations. First, it should be noted that varying study designs, including study population, sample size, hospital setting, infection type, and hospital ward could influence the clinical and economic outcomes. However, most of included studies did not differentiate which of these culture results represented true infection or colonization. Colonization, as an important reservoir for strains causing healthcare-associated infections, should be considered in future research. In addition, only study one was prospective, and the nature of retrospective studies means they may result in missing data and selection bias. Only published literatures were included, and potential publication bias cannot be neglected. Lastly, we could not conduct a meta-analysis for hospital stay and hospital costs due to a variety of reporting values (mean or median). 

## 5. Conclusions

Our study indicates that ABR is associated with significantly higher mortality, whether in unadjusted or adjusted analyses. Moreover, ABR is not always, but usually, associated with significantly longer hospital stay and higher hospital costs. It is possible to lack statistical power to detect significant differences; however, the results without adjustments for confounding factors need to be interpreted with caution. The review also highlights key areas where further research is needed in China: there is a need for prospective studies with multiple settings, with a societal perspective, and large sample size. In addition, a standardized and localized definition about ABR or MDR is necessary in China. Research is needed in the future, focusing on other bacteria (e.g., *Enterococcus*, *E. coli*) and colonized bacteria as well.

## Figures and Tables

**Figure 1 antibiotics-08-00115-f001:**
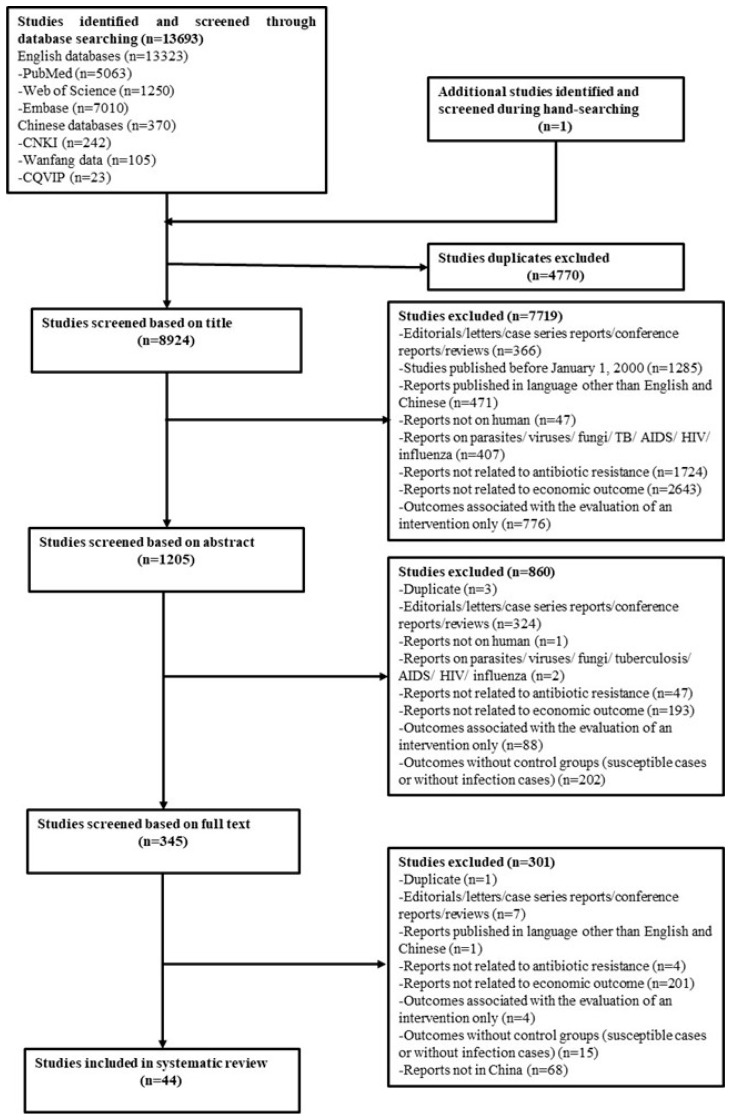
Flowchart of literature search. CNKI: China National Knowledge Infrastructure; CQVIP: Chongqing VIP; TB: Tuberculosis; AIDS: acquired immunodeficiency syndrome; HIV: human immunodeficiency virus.

**Figure 2 antibiotics-08-00115-f002:**
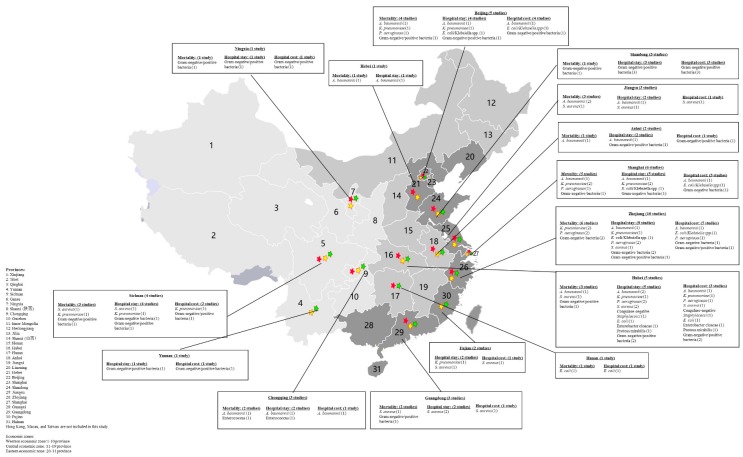
Graphical representation of antibiotic resistance in mainland China in this study. *S. aureus*: *Staphylococcus aureus*; *K. pneumoniae*: *Klebsiella pneumoniae*; *A. baumannii*: *Acinetobacter baumannii*; *P. aeruginosa*: *Pseudomonas aeruginosa*; *E. coli*: *Escherichia coli*.

**Figure 3 antibiotics-08-00115-f003:**
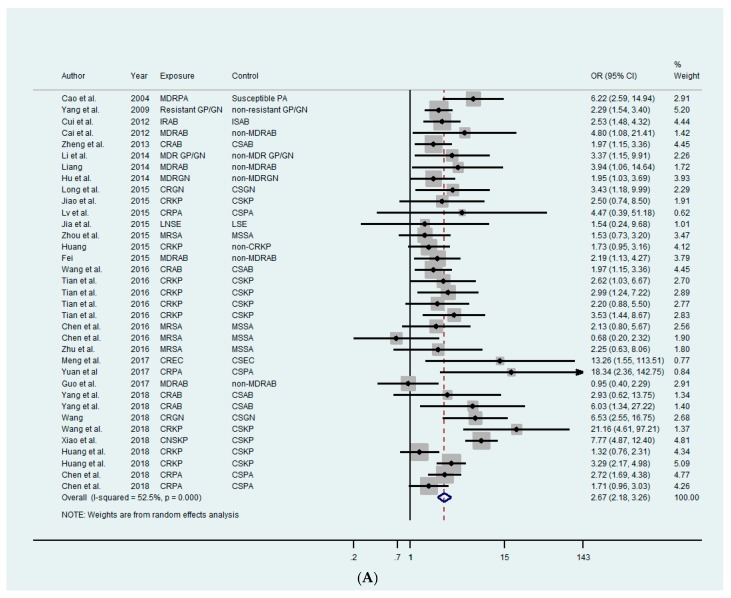
Forest plot of impact of antibiotic resistance on mortality and sub-group analyses. (**A**) Forest plot of overall mortality of antibiotic resistance compared to those with susceptibility. (**B**) Forest plot of overall mortality of antibiotic resistance compared to those without infection. (**C**) Forest plot of overall mortality of antibiotic resistance compared to those with susceptibility based on bacteria. (**D**) Forest plot of overall mortality of antibiotic resistance compared to those with susceptibility based on economic zones (eastern economic zone, central economic zone, and western economic zone). OR: odds ratio; CI: confidence intervals; PA: *Pseudomonas aeruginosa*; MDRPA: multi-drug resistant *P. aeruginosa*; GP/GN: gram-positive/negative bacteria; IRAB: imipenem-resistant *Acinetobacter baumannii*; ISAB: imipenem-susceptible *A. baumannii*; MDRAB: multi-drug resistant *A. baumannii*; CRAB: carbapenem-resistant *A. baumannii*; CSAB: carbapenem-susceptible *A. baumannii*; MDR GP/GN: multi-drug resistant gram-positive/negative bacteria; CRGN: carbapenem-resistant gram-negative bacteria; CSGN: carbapenem-susceptible gram-negative bacteria; CRKP: carbapenem-resistant *Klebsiella pneumoniae*; CSKP: carbapenem-susceptible *K. pneumoniae*; CRPA: carbapenem-resistant *P. aeruginosa*; CSPA: carbapenem-susceptible *P. aeruginosa*; LNSE: linezolid non-susceptible *Enterococcus*; LSE: linezolid-susceptible *Enterococcus*; MRSA: methicillin-resistant *Staphylococcus aureus*; MSSA: methicillin-susceptible *S. aureus*; CREC: carbapenem-resistant *Escherichia coli*; CSEC: carbapenem-susceptible *E. coli*; CNSKP: carbapenem non-susceptible *K. pneumoniae*; MDRGN: multi-drug resistant gram-negative bacteria.

**Table 1 antibiotics-08-00115-t001:** Studies describing hospital mortality among inpatients with antibiotic resistance and multi-drug resistance.

Author	Year	Bacteria	Comparison Group		Sample Size		Description of Mortality	Mortality Rate				*p*-Value
			Case	Control	Case	Control		Case		Control		
								*n*	%	*n*	%	
Guo et al. [27]	2017	*A. baumannii*	MDR	non-MDR	122	366	in hospital mortality	7	5.74	22	6.01	0.912
Hu et al. [28]	2014	gram-negative	MDR	non-MDR	89	165	30-day hospital mortality	23	25.8	25	15.2	<0.05
Huang [29]	2015	*K. pneumoniae*	CRKP	non-CRKP	113	77	in hospital mortality	53	46.9	26	33.77	0.07
Li et al. [30]	2014	gram-negative/gram positive	MDR	non-MDR	62	473	in hospital mortality	5	8.07	12	2.54	<0.05
Liang [31]	2014	*A. baumannii*	MDR	non-MDR	68	53	in hospital mortality	13	19.12	3	5.66	0.03
Lv et al. [32]	2015	*P. aeruginosa*	CRPA	CSPA	32	68	in hospital mortality	2	13.33	1	1.79	<0.05
Pei [33]	2015	*A. baumannii*	MDR	non-MDR	226	65	in hospital mortality	80	35.4	13	20	0.019
Wang [34]	2018	gram-negative	carbapenem resistance	carbapenem susceptibility	26	113	28-day hospital mortality	13	50	15	13.3	<0.001
Wang et al. [35]	2016	*A. baumannii*	CRAB	CSAB	97	145	in hospital mortality	44	45.6	43	29.9	0.02
Zhang et al. [36]	2013	*S. aureus*	MRSA	without infection	192	384	in hospital mortality	21	10.94	17	4.43	0.03
Zhou et al. [37]	2015	*S. aureus*	MRSA	MSSA	91	266	in hospital mortality	12	13.19	24	9.02	0.26
Chen et al. [38]	2016	*S. aureus*	MRSA	MSSA	75	78	in hospital mortality	13	17.33	7	8.97	0.131
					46	46	in hospital mortality	5	10.87	7	15.22	0.385
Cui et al. [39]	2012	*A. baumannii*	IRAB	ISAB	138	138	in hospital mortality	54	39.1	28	20.3	<0.01
Long et al. [40]	2015	Gram-negative	carbapenem resistance	carbapenem susceptibility	34	34	in hospital mortality	16	47.1	7	20.6	0.021
Zhu et al. [41]	2016	*S. aureus*	MRSA	MSSA	22	42	in hospital mortality	6	27.3	6	14.3	0.312
Yang et al. [42]	2018	*A. baumannii*	CRAB	CSAB	84	34	in hospital mortality	23	27.4	2	5.9	0.011
					84	34	30-day hospital mortality	13	15.5	2	5.9	0.025
Chen et al. [43]	2018	*P. aeruginosa*	CRPA	CSPA	327	472	in hospital mortality	51	15.6	30	6.4	<0.001
					270	270	in hospital mortality	34	12.6	21	7.8	0.044
Meng et al. [44]	2017	*E. coli*	CREC	CSEC	49	96	in hospital mortality	6	12	1	1	0.01
			CREC	without infection	49	96	in hospital mortality	6	12	1	1	0.01
Zheng et al. [45]	2013	*A. baumannii*	CRAB	CSAB	97	145	28-day hospital mortality	44	45.6	43	29.9	0.02
Yuan et al. [46]	2017	*P. aeruginosa*	CRPA	CSPA	85	94	in hospital mortality	14	16.5	1	1.1	<0.001
Xiao et al. [47]	2018	*K. pneumoniae*	CNSKP	CSKP	135	293	30-day hospital mortality	79	58.5	45	15.4	<0.001
Wang et al. [48]	2018	*K. pneumoniae*	CRKP	CSKP	48	48	in hospital mortality	23	47.9	2	4.2	0.03
Tian et al. [49]	2016	*K. pneumoniae*	CRKP	CSKP	33	81	in hospital mortality	14	42.4	16	19.8	0.013
					33	81	28-day hospital mortality	11	33.3	15	18.5	0.087
					33	81	attributable 28-day hospital mortality	11	33.3	13	16	0.04
					33	81	attributable in hospital mortality	14	42.4	14	24.6	0.005
Jiao et al. [50]	2015	*K. pneumoniae*	CRKP	CSKP	30	30	in hospital mortality	10	33.3	5	16.7	>0.05
Huang et al. [51]	2018	*K. pneumoniae*	CRKP	CSKP	237	237	in hospital mortality	32	13.5	25	10.55	0.324
					237	1328	in hospital mortality	39	14.61	75	5.65	<0.001
Yang et al. [52]	2009	gram-positive/gram-negative	resistance	non-resistance	676	732	in hospital mortality	79	11.7	40	5.4	<0.001
Cao et al. [53]	2004	*P. aeruginosa*	MDR *P. aeruginosa*	susceptibility	44	68	in hospital mortality	24	54.5	11	16.2	<0.05
Jia et al. [54]	2015	*Enterococcus*	linezolid non-susceptibility	linezolid susceptibility	44	44	in hospital mortality	3	6.8	2	4.5	>0.05
			linezolid non-susceptibility	Inpatients during the same time	44	176	in hospital mortality	3	6.8	3	1.7	>0.05
Cai et al. [55]	2012	*A. baumannii*	MDR	non-MDR	115	45	in hospital mortality	21	18.26	2	4.44	<0.05

*A. baumannii*: *Acinetobacter baumannii*; *K. pneumoniae*: *Klebsiella pneumoniae*; *P. aeruginosa*: *Pseudomonas aeruginosa*; *S. aureus*: *Staphylococcus aureus*; *E. coli*: *Escherichia coli*; MDR: multi-drug resistance; CRKP: carbapenem-resistant *K. pneumoniae*; CSKP: carbapenem-susceptible *K. pneumoniae*; CRPA: carbapenem-resistant *P. aeruginosa*; CSPA: carbapenem-susceptible *P. aeruginosa*; CRAB: carbapenem-resistant *A. baumannii*; CSAB: carbapenem-susceptible *A. baumannii*; IRAB: imipenem-resistant *A. baumannii*; ISAB: imipenem-susceptible *A. baumannii*; MRSA: methicillin-resistant *S. aureus*; MSSA: methicillin-susceptible *S. aureus*; CREC: carbapenem-resistant *E. coli*; CSEC: carbapenem-susceptible *E. coli*; CNSKP: carbapenem non-susceptible *K. pneumoniae*.

**Table 2 antibiotics-08-00115-t002:** Studies describing hospital stay among patients with antibiotic resistance and multi-drug resistance.

Author	Year	Bacteria	Comparison Group		Sample Size		Description of LOS		LOS				*p*-Value
			Case	Control	Case	Control			Case		Control		
Fu et al. [56]	2014	*S. aureus*	MRSA	without infection	456	706	total LOS	median (Q)	31	42	16	14	0.001
Guo et al. [27]	2017	*A. baumannii*	MDR	non-MDR	122	366	total LOS	mean (SD)	24	17	11	9	<0.001
								median (Q1-Q3)	19	(13–29)	9	(5–15)	<0.001
Hu et al. [28]	2014	gram-negative	MDR	non-MDR	89	165	total LOS	median (IQR)	24	(18–39)	25	(17–52)	>0.05
Huang [29]	2015	*K. pneumoniae*	CRKP	non-CRKP	113	77	total LOS	mean (SD)	70	69	32	34	<0.000
							ICU LOS	mean (SD)	59	70	22	33	<0.001
Jiang et al. [57]	2016	gram-negative/gram-positive	MDR	non-MDR	41	41	total LOS	median (Q)	24	25	19	14	0.01
Li et al. [58]	2018	gram-negative/gram-positive	MDR	susceptibility	78	78	total LOS	mean (SD)	14	6	11	3	<0.001
Li et al. [59]	2016	*K. pneumoniae*	resistant enzymes producing	non-resistant enzymes producing	41	80	total LOS	mean (SD)	22	17	14	9	0.003
							LOS before infection	mean (SD)	8	8	5	5	0.017
Liang [31]	2014	*A. baumannii*	MDR	non-MDR	68	53	total LOS	mean (SD)	24	10	14	5	0.002
Liu [60]	2018	gram-negative/gram-positive	antibiotic resistance	without nosocomial infection	133	133	total LOS	mean	68		28		<0.05
Lv et al. [32]	2015	*P. aeruginosa*	CRPA	CSPA	32	68	LOS after admitting ICU	mean (SD)	11	1	3	1	0.01
Pan et al. [61]	2018	gram-negative/gram-positive	MDR	susceptibility	102	79	total LOS	mean (SD)	36	22	29	18	0.026
Pei [33]	2015	*A. baumannii*	MDR	non-MDR	226	65	total LOS	mean (SD)	35	25	27	17	0.002
Wang [34]	2018	gram-negative	carbapenem resistance	carbapenem susceptibility	26	113	LOS before infection	median (IQR)	19	(3–42)	3	(0–13)	<0.001
Jiang [62]	2018	gram-negative	MDR	non-MDR	79	79	total LOS	mean (SD)	19	15	13	7	<0.05
							LOS before infection	mean (SD)	10	5	9	7	<0.05
Wang et al. [35]	2016	*A. baumannii*	CRAB	CSAB	97	145	LOS before pneumonia	mean (SD)	18	6	18	7	0.38
Wu et al. [63]	2018	*A. baumannii*	MDR	non-MDR	65	65	total LOS	median (Q)	52	42	27	21	<0.01
Xing et al. [64]	2017	gram-negative/gram-positive	MDR	without infection	178	178	total LOS	median (IQR)	32	(23–47)	12	(9–27)	<0.001
Xu et al. [65]	2017	*E. coli*	MDR	non-MDR	969	1940	total LOS	mean (SD)	19	23	13	12	<0.001
		*K. pneumoniae*	MDR	non-MDR	186	529	total LOS	mean (SD)	19	16	15	14	0.03
		*Proteus mirabilis*	MDR	non-MDR	38	60	total LOS	mean (SD)	25	22	14	9	0.002
		*A. baumannii*	MDR	non-MDR	53	45	total LOS	mean (SD)	22	21	16	11	0.045
		*P. aeruginosa*	MDR	non-MDR	13	490	total LOS	mean (SD)	64	43	18	17	<0.001
		*Enterobacter cloacae*	MDR	non-MDR	94	166	total LOS	mean (SD)	29	31	18	19	0.001
		*S. aureus*	MDR	non-MDR	41	237	total LOS	mean (SD)	21	18	14	15	0.008
		coagulase-negative *Staphylococci*	MDR	non-MDR	76	184	total LOS	mean (SD)	26	26	18	16	0.002
Yu [66]	2016	*S. aureus*	MRSA	MSSA	118	116	total LOS	median	33		14		<0.05
Zhang et al. [36]	2013	*S. aureus*	MRSA	without infection	192	384	total LOS	mean (SD)	27	9	18	9	<0.01
Zhou et al. [37]	2015	*S. aureus*	MRSA	MSSA	91	266	total LOS	median (IQR)	29	(21–60)	23	(15–42)	<0.01
							LOS before infection	median (IQR)	11	(4–23)	3.5	(0–13)	<0.01
							LOS after infection	median (IQR)	17	(7–31)	16.5	(8–29)	0.92
Chen et al. [38]	2016	*S. aureus*	MRSA	MSSA	75	78	total LOS	median (IQR)	40	(20–94)	28	(21–53)	0.003
					46	46	total LOS	median (IQR)	28	(21–52)	28	(21–53)	0.899
					75	78	LOS after infection	median (IQR)	19	(10–46)	17	(8–29)	0.011
					46	46	LOS after infection	median (IQR)	15	(9–25)	17	(8–29)	0.676
Cui et al. [39]	2012	*A. baumannii*	IRAB	ISAB	138	138	total LOS	median (IQR)	29	(19–57)	23	(15–39)	<0.01
							ICU LOS	median (IQR)	15	(8–28)	0	(0–10)	<0.01
							LOS before infection	median (IQR)	10	(4–20)	13	(7–20)	>0.05
Long et al. [40]	2015	gram-negative	carbapenem resistance	carbapenem susceptibility	34	34	total LOS	mean (SD)	28	3	22	2	>0.05
							ICU LOS	mean (SD)	17	3	13	3	>0.05
Zhu et al. [41]	2016	*S. aureus*	MRSA	MSSA	22	42	total LOS	mean (SD)	26	23	15	11	0.062
Hu et al. [67]	2010	*E. coli*/*Klebsiella spp.*	ESBL-positive	ESBL-negative	32	53	total LOS	mean	24		15		0.001
Zhen et al. [68]	2017	*A. baumannii*	CRAB	CSAB	2126	854	LOS before infection	mean (SD)	10	16	11	28	0.057
Zhen et al. [69]	2018	gram-negative/gram-positive	MDR	non-MDR	64	37	total LOS	mean (SD)	31	29	16	13	<0.000
Chen et al. [43]	2018	*P. aeruginosa*	CRPA	CSPA	327	472	total LOS	median (IQR)	29	(17–44)	21	(11–34)	<0.001
					270	270	total LOS	median (IQR)	29	(17–42)	26	(14–41)	0.026
					327	472	LOS after infection	median (IQR)	17	(8–32)	13	(7–25)	0.005
					270	270	LOS after infection	median (IQR)	19	(8–30)	14	(7–28)	0.029
Wang et al. [48]	2018	*K. pneumoniae*	CRKP	CSKP	48	48	total LOS	median (IQR)	84	(41–188)	33	(21–60)	0.097
Tian et al. [49]	2016	*K. pneumoniae*	CRKP	CSKP	33	81	total LOS	median (IQR)	50	(28–83)	24	(16.5–51)	0.001
							LOS after infection	median (IQR)	24	(10–51)	15	(9–28)	0.066
Jiao et al. [50]	2015	*K. pneumoniae*	CRKP	CSKP	30	30	total LOS	mean (SD)	34	31	18	23	0.054
							LOS before infection	mean (SD)	34	31	13	27	0.02
Huang et al. [51]	2018	*K. pneumoniae*	CRKP	CSKP	237	237	total LOS	median (range)	31	(22–55)	24	(14–46)	<0.001
					237	1328	total LOS	median (range)	31	(22–56)	19	(11–35)	<0.001
					237	1328	LOS before infection	median (range)	13	(2–25)	3	(0–11)	<0.001
					237	1328	LOS after infection	median (range)	21	(10–44)	18	(9–46)	0.612
Yang et al. [52]	2009	gram-negative/gram-positive	resistance	non-resistance	676	732	total LOS	mean (SD)	34	39	18	24	<0.001
							total LOS	median	21		12		<0.001
							infection related LOS	mean (SD)	22	21	12	13	<0.001
							infection related LOS	median	15		9		<0.001
Li et al. [70]	2016	*S. aureus*	MRSA	MSSA	14	61	total LOS	mean (SD)	38	47	19	14	0.12
							total LOS	median	19		15		0.12
Jia et al. [54]	2015	*Enterococcus*	linezolid nonsusceptibility	linezolid susceptibility	44	44	total LOS	median (IQR)	37	(15–57)	22	(9–43)	<0.05
			linezolid nonsusceptibility	inpatients during the same time	44	176	total LOS	median (IQR)	37	(15–57)	17	(11–28)	<0.05
			linezolid nonsusceptibility	linezolid susceptibility	44	44	LOS after infection	median (IQR)	8	(3–15)	5	(3–20)	<0.05
			linezolid nonsusceptibility	inpatients in the same time	44	176	LOS after infection	median (IQR)	8	(3–15)	4	(1–12)	<0.05
Cai et al. [55]	2012	*A. baumannii*	MDR	non-MDR	115	45	total LOS	mean (SD)	19	9	14	4	0.001
							ICU LOS	mean (SD)	17	7	14	4	0.009

*S. aureus*: *Staphylococcus aureus*; *A. baumannii*: *Acinetobacter baumannii*; *K. pneumoniae*: *Klebsiella pneumoniae*; *P. aeruginosa*: *Pseudomonas aeruginosa*; *E. coli*: *Escherichia coli*; MRSA: methicillin-resistant *S. aureus*; MSSA: methicillin-susceptible *S. aureus*; MDR: multi-drug resistance; CRKP: carbapenem-resistant *K. pneumoniae*; CSKP: carbapenem-susceptible *K. pneumoniae*; CRPA: carbapenem-resistant *P. aeruginosa*; CSPA: carbapenem-susceptible *P. seruginosa*; CRAB: carbapenem-resistant *A. baumannii*; CSAB: carbapenem-susceptible *A. baumannii*; IRAB: imipenem-resistant *A. baumannii*; ISAB: imipenem-susceptible *A. baumannii*; ESBL: extended spectrum βlactamases; ICU: intensive care unit; LOS: length of stay; SD: standard deviation; IQR: interquartile range; Q: quartile.

**Table 3 antibiotics-08-00115-t003:** Studies describing hospital costs among patients with antibiotic resistance and multi-drug resistance.

Author	Year	Bacteria	Comparison Group		Sample Size		Description of Cost	Mean (Median) Costs in 2015 USD		*p*-Value
			Case	Control	Case	Control		Case	Control	
Fu et al. [56]	2014	*S. aureus*	MRSA	without infection	456	706	total hospital cost	(15,763)	(2185)	0.001
Li et al. [70]	2016	*S. aureus*	MRSA	MSSA	14	61	total hospital cost	5305(319)	2658(352)	0.39
Chen et al. [38]	2016	*S. aureus*	MRSA	MSSA	75	78	treatment cost	(23,933)	(19,905)	0.395
					46	46	treatment cost	(19,718)	(19,538)	0.935
Hu et al. [28]	2014	gram-negative	MDR	non-MDR	89	165	total hospital cost	(12,360)	(11,591)	>0.05
					89	165	antibiotic cost	(1946)	(1397)	<0.01
Long et al. [40]	2015	gram-negative	carbapenem resistance	carbapenem susceptibility	34	34	total treatment cost	11,206	6686	0.034
Jiang et al. [57]	2016	gram-positive/gram-negative	MDR	non-MDR	41	41	total hospital cost	(10,832)	(6607)	<0.00
Li et al. [58]	2018	gram-positive/gram-negative	MDR	susceptibility	78	78	total hospital cost	1660	1093	<0.001
					78	78	antibiotic cost	485	322	<0.001
Liu [60]	2018	gram-positive/gram-negative	antibiotic resistance	without nosocomial infection	133	133	total hospital cost	20,222	3726	<0.05
Pan et al. [61]	2018	gram-positive/gram-negative	MDR	susceptibility	102	79	total hospital cost	12,602	9793	<0.001
					102	79	antibiotic cost	952	740	<0.001
Yang et al. [52]	2009	gram-positive/gram-negative	resistance	non-resistance	676	732	total hospital cost	11,035(4303)	2940(1103)	<0.001
					676	732	antibiotic cost	812(418)	274(119)	<0.000
Xing et al. [64]	2017	gram-positive/gram-negative	MDR	without infection	178	178	total hospital cost	(16,138)	(1714)	<0.001
Zhen et al. [69]	2018	gram-positive/gram-negative	MDR	non-MDR	64	37	total hospital cost	21,164	6680	<0.000
					64	37	antibiotic cost	4001	760	<0.000
Guo et al. [27]	2017	*A. baumannii*	MDR	non-MDR	122	366	total hospital cost	14,159(10,452)	7487(3759)	<0.001
Wu et al. [63]	2018	*A. baumannii*	MDR	non-MDR	65	65	total hospital cost	(24,897)	(8823)	<0.01
					65	65	daily hospital cost	(581)	(688)	0.14
Cui et al. [39]	2012	*A. baumannii*	IRAB	ISAB	138	138	daily total hospital cost	(591)	(338)	<0.01
					138	138	daily antibiotic cost	(90)	(55)	<0.01
Zhen et al. [68]	2017	*A. baumannii*	CRAB	CSAB	2126	854	total hospital cost	30,575	19,783	<0.000
					2126	854	antibiotic cost	3047	1692	<0.000
Chen et al. [43]	2018	*P. aeruginosa*	CRPA	CSPA	327	472	total hospital cost	(925)	(482)	<0.001
					270	270	total hospital cost	(868)	(707)	0.015
					327	472	daily hospital cost	(36)	(27)	<0.001
					270	270	daily hospital cost	(34)	(32)	0.045
Xu et al. [65]	2017	*E. coli*	MDR	non-MDR	969	1940	total hospital cost	3645	2071	<0.001
					969	1940	antibiotic cost	234	154	<0.001
		*K. pneumoniae*	MDR	non-MDR	186	529	total hospital cost	5132	3178	0.001
					186	529	antibiotic cost	263	246	0.59
		*Proteus mirabilis*	MDR	non-MDR	38	60	total hospital cost	6383	2700	<0.001
					38	60	antibiotic cost	271	114	0.001
		*A. baumannii*	MDR	non-MDR	53	45	total hospital cost	5446	3100	0.025
					53	45	antibiotic cost	222	136	0.054
		*P. aeruginosa*	MDR	non-MDR	13	490	total hospital cost	13,820	3847	<0.001
					13	490	antibiotic cost	884	325	<0.001
		*Enterobacter cloacae*	MDR	non-MDR	94	166	total hospital cost	7788	3812	<0.001
					94	166	antibiotic cost	386	255	0.01
		*S. aureus*	MDR	non-MDR	41	237	total hospital cost	4139	2355	0.006
					41	237	antibiotic cost	223	141	0.007
		coagulase-negative *Staphylococci*	MDR	non-MDR	76	184	total hospital cost	9028	3215	<0.001
					76	184	antibiotic cost	362	212	<0.001
Hu et al. [67]	2010	*E. coli/Klebsiella spp*.	ESBL-positive	ESBL-negative	32	53	total hospital cost	541	303	<0.001
					32	53	cost of intravenous antibiotics	98	40	0.001
Meng et al. [44]	2017	*E. coli*	CREC	CSEC	49	96	total hospital cost	(12,670)	(10,290)	0.05
				without infection	49	96	total hospital cost	(12,670)	(2818)	<0.00
Huang et al. [51]	2018	*K. pneumoniae*	CRKP	CSKP	237	237	total hospital cost	(21,170)	(11,313)	<0.001
					237	237	total antibiotic cost	(2253)	(1251)	<0.01
					237	237	hospital cost after infection	(8912)	(6677)	0.003
					237	237	antibiotic cost after infection	(973)	(573)	<0.001

*S. aureus*: *Staphylococcus aureus*; *A. baumannii*: *Acinetobacter baumannii*; *P. aeruginosa*: *Pseudomonas aeruginosa*; *K. pneumoniae*: *Klebsiella pneumoniae*; *E. coli*: *Escherichia coli*; MRSA: methicillin-resistant *S. aureus*; MSSA: methicillin-susceptible *S. aureus*; MDR: multi-drug resistance; CRKP: carbapenem-resistant *K. pneumoniae*; CSKP: carbapenem-susceptible *K. pneumoniae*; IRAB: imipenem-resistant *A. baumannii*; ISAB: imipenem-susceptible *A. baumannii*; CRAB: carbapenem-resistant *A. baumannii*; CSAB: carbapenem-susceptible *A. baumannii*; CRPA: carbapenem-resistant *P. aeruginosa*; CSPA: carbapenem-susceptible *P. seruginosa*; ESBL: extended spectrum βlactamases; CREC: carbapenem-resistant *E. coli*; CSEC: carbapenem-susceptible *E. coli*; USD: United States Dollars.

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
