# Peer review of "The Clinical and Economic Impact of Antibiotic Resistance in China: A Systematic Review and Meta-Analysis"

_antibiotics, 2019, doi:10.3390/antibiotics8030115_

Round 1

Reviewer 1 Report

The review provides important information for readers interested in the global effects of antibiotic use. The Chinese language databases have been searched making dissemination useful. Despite strict government interventions use has increased - this needs to be explained. How is multidrug resistance defined for this review? 

Line 134 does 'significant tests' mean statistical significance? If so what was compared? Missing data could affect data extraction results - was missing data random?

Line 203 Klebsiella pneumoniae, note spelling. Where multiple significant results were obtained was any allowance made for random events? Were adverse events from broader spectrum antibiotics used to treat MDR infections a major issue or was it less effective treatment that had the effect on costs and length of stay?

Discussion - the reasons and extent of differences in antibiotic use between China and the rest of the world need to be described fully - for instance the heavier use of intravenous agents. What is the level of control and acceptance of stewardship in prescription of antibiotics?

Author Response

Dear Reviewer,

Many thanks for the invaluable comments. We have revised the manuscript based on the comments. The following shows a point-by-point response to the comments, we repeat each comment in bold and give the response in plain in text.

Comments and Suggestions for Authors:

The review provides important information for readers interested in the global effects of antibiotic use. The Chinese language databases have been searched making dissemination useful.

1. Despite strict government interventions use has increased - this needs to be explained. 

Response: We have revised the manuscript accordingly.

Strict government interventions were conducted to combat the problem of antibiotic use and antibiotic resistance in China, however, the problem is still acute in China, which may related to its antibiotic prescribing behavior, including: inappropriate financial incentive, over-the-counter availability of antibiotics, and the widespread antibiotic use and misuse in agriculture.

In addition, it was reported that China is one of the world’s largest producers and consumers of antibiotics. However, the measures for combatting the problem of antibiotic use and antibiotic resistance in China started from 2011. Therefore, the effect of interventions are effective but limited. It needs more time to see the better results.

In the revised manuscript, we explained that we faced a great challenge in controlling antibiotic use and antibiotic resistance in China, even though, a series of measures were conducted.

Please see in Introduction section, lines 28-58, pages 1-2.

2. How is multidrug resistance defined for this review? 

Response: We have defined multidrug resistance in the revised manuscript

Multi-drug resistance was defined that if it is resistance to three or more than three types of antibiotics or if the isolated bacteria were MDR organisms, such as MRSA, CRPA, CRAB.

Please see the Materials and Methods section, lines 94-96, page 3.

3. Line 134 does 'significant tests' mean statistical significance? If so what was compared? Missing data could affect data extraction results - was missing data random?

Response: We are so sorry for our careless.

The “significant tests” means statistical test, which is one of the most utilized methods to compare the difference in clinical and economic outcome between antibiotic resistant or multi-drug resistant group and susceptible group or uninfected group.

We try to extract all information from included studies according to our objectives. In the included studies, only one study was prospective, and the nature of retrospective studies may result in missing data and selection bias. We have discussed the missing data in the limitations.

Please see in Materials and Methods section, lines 88-96, pages 2-3; Results section, line 136, page 4 Discussion section, lines 354-355, page 21.

4. Line 203 Klebsiella pneumoniae, note spelling. Where multiple significant results were obtained was any allowance made for random events? Were adverse events from broader spectrum antibiotics used to treat MDR infections a major issue or was it less effective treatment that had the effect on costs and length of stay?

Response: We are so sorry for our careless. We have revised all spelling mistakes of K. pneumoniae in the revised manuscript.

All included studies were observational studies. There were multiple significant results in one study because of different comparative groups, different analyzed methods, different reporting values (mean or median) or different types of cost (antibiotic cost, total hospital charge, or total hospital cost).

Adverse events from broader spectrum antibiotics used to treat MDR infection were major issues, and less effective treatment have important effect on costs and length of stay. Both adverse events from broader spectrum antibiotics and less effective treatment can contributed to higher mortality rate, longer length of stay, and higher hospital costs. However, we cannot conclude which one is more important from this systematic review. We can investigate it in the future.

5. Discussion - the reasons and extent of differences in antibiotic use between China and the rest of the world need to be described fully - for instance the heavier use of intravenous agents. What is the level of control and acceptance of stewardship in prescription of antibiotics?

Response: Thank you very much for your comments.

In the Discussion, we reported that ABR is a global health crisis, especially in China, with high prescription rates for antibiotics in both inpatients and outpatients, coupled with the highest level growth of ABR globally, at least until recently.

As we know, antibiotic use and misuse can contribute to the developing of antibiotic resistance. China is one of the world’s largest producers and consumers of antibiotics, which may be related to its antibiotic prescribing behavior, including: inappropriate financial incentives, over-the-counter availability of antibiotics, and the widespread antibiotic use and misuse in agriculture.

It was reported that there is also a high use of injections in China, about one-third of the prescriptions were for injections in community health institutions, which is two to three times higher than the WHO standard and estimates from other developing countries.

Chinese governments conducted a series of measures to combat antibiotic resistance, especially multi-drug resistance. Chinese government announced a national action plan to combat antimicrobial resistance in 2016 in response to the global action plan by WHO. On July 1, 2011, the Chinese government carried out a three-year national public hospital campaign targeting ABR. This action plan, as a combination of managerial and professional strategies, was effective in reducing antibiotic prescribing rates and intensity of antibiotic consumption. On August 1, 2012, the Chinese government formally implemented administrative regulations for the clinical use of antibacterial agents. In addition, China has built multi-disciplinary collaborations with the European Union, Sweden, the Netherlands, and the United Kingdom to stop the increasing the burden caused by ABR.

Finally, we found that except the isolated rates of CRKP and CRAB, the rates of other multi-drug resistant bacteria, such as MRSA, CRPA, ESBL- E. coli, ESBL-K. pneumoniae showed downward trends. It highlights that the measures are effective, and may need longer time to see the better results.

Please see in Introduction section, lines 28-58, page 1-2; Discussion section, lines 286-291, page 20.

Reviewer 2 Report

In general a piece of work with good quality, and indeed enriched the literature in the burden of AMR globally and in LMICs.

Please add the grey literature sources that you have screened - if you, please provide justification.

  1. In the study inclusion section, age group and gender are not specified. Please state studies in human of all ages and all sexes were included if that is the case.   2. Please state the consideration for studies that focused on both humans and animals.   3. Author only included methods of cohort studies, case control studies and observational studies. Observational studies does seem to be clear. Also, please explain why cross-sectional studies, longitudinal studies, clinical trials, economic evaluations, and modelling studies are not included, especially economic evaluations and clinical trials.    4. Please explain why grey literature was not searched.   5. The way of presenting included studies using a map is excellent. Please also expand the discussion in the geographical heterogeneity of the issue (AMR burden) in China.   6. in the manuscript, the author did not provide information in intermediate isolates. There is a growing body of literature in research of AMR suggesting intermediate isolates should be treated separately, instead of being clustered as resistant. Please explain how intermediate isolates were analysed in this study.    7. No established quality assessment was used in this study, which compromised the credibility of the result reported. Please consider using a framework such as PICO to examine the quality of the studies included and report potential bias, especially the screening was done in two languages.    8. The studies reported only clinical burden, or economic burden should be included for screening too.   In this study a literature review conducted in order to determine the burden of AMR in China, overall it is a good paper that provides information for future research priority setting and policy design.   

Author Response

Dear Reviewer,

Many thanks for the invaluable comments. We have revised the manuscript based on the comments. The following shows a point-by-point response to the comments, we repeat each comment in bold and give the response in plain in text.

Comments and Suggestions for Authors:

In general a piece of work with good quality, and indeed enriched the literature in the burden of AMR globally and in LMICs. In this study a literature review conducted in order to determine the burden of AMR in China, overall it is a good paper that provides information for future research priority setting and policy design.  

Please add the grey literature sources that you have screened-if you, please provide justification.

Response: In this study, we excluded grey literature sources and only included original articles.

1. In the study inclusion section, age group and gender are not specified. Please state studies in human of all ages and all sexes were included if that is the case.  

Response: In the inclusion criteria, age and gender are not specified.

In addition, sex was regarded as an influencing factor in all included studies, and some studies only included adults or children as study population.

Presenting all information in the Table makes confused, so we present some information in supplementary data.

Please see the characteristic-Study Population in Table S1, S2 S3 in Supplementary data 3.

2. Please state the consideration for studies that focused on both humans and animals.  

Response: In this study, we only consider for studies that focused on humans.

Please see in Materials and Methods section, line 81, page 2.

3. Author only included methods of cohort studies, case control studies and observational studies. Observational studies does seem to be clear. Also, please explain why cross-sectional studies, longitudinal studies, clinical trials, economic evaluations, and modelling studies are not included, especially economic evaluations and clinical trials.   

Response: We are so sorry for your mistakes. We have revised the manuscript accordingly.

In the study selection, we included all original articles with any study designs. Finally, we found all included studies were observational studies.

Please see in Materials and Methods section, lines 80-81, page 2.

4. Please explain why grey literature was not searched.  

Response: First, we would like to analyze the clinical and economic consequences of antibiotic resistant or multidrug resistant bacteria compared to susceptible bacteria and uninfected individuals. There existed some missing data influencing data extraction results in grey literature.

In addition, we also discuss the publication bias in the limitations.

Please see in Discussion section, lines 355-356, page 21.

5. The way of presenting included studies using a map is excellent. Please also expand the discussion in the geographical heterogeneity of the issue (AMR burden) in China.  

Response: We have revised the manuscript accordingly.

There was geographical heterogeneity of studies reporting on clinical and economic outcomes in China. The most studies are limited chiefly to eastern economic zone, which is the most developed zone in China. Its consistent with the situation that similar analyses are needed for low- and middle- income countries. The current status of ABR or MDR may be more serious in central and western economic zone because of lacking of new medicines, diagnostic tools, and interventions. Moreover, compared with eastern economic zone, ABR or MDR in central and western economic zone may be associated with higher mortality rate and higher economic burden, and more likely to be forced into extreme poverty

Please see in Discussion section, lines 340-348, page 21.

6. In the manuscript, the author did not provide information in intermediate isolates. There is a growing body of literature in research of AMR suggesting intermediate isolates should be treated separately, instead of being clustered as resistant. Please explain how intermediate isolates were analysed in this study.   

Response: We agree with the reviewer’s comment that there is a growing body of literature in research of AMR suggesting intermediate isolates should be treated separately, instead of being clustered as resistant. However, in this study, intermediate susceptibility was considered as resistance according to the definitions in included studies.

Please see in Materials and Methods section, line 96, page 3.

7. No established quality assessment was used in this study, which compromised the credibility of the result reported. Please consider using a framework such as PICO to examine the quality of the studies included and report potential bias, especially the screening was done in two languages.   

Response: We assessed the included study quality using the Newcastle-Ottawa quality assessment Scale (NOS) for cohort and case-control studies.

Please see in Materials and Methods section, lines 97-101, page 3; and Table S4 and S5 in Supplementary data 2.

8. The studies reported only clinical burden, or economic burden should be included for screening too.  

Response: We reported both clinical burden and economic burden in this study.

Please see in Results section, lines 143-264, pages 5-10; Table 1-3, and Table S1-S3 in Supplementary data 3.